



# Supercooled liquid water and secondary ice production in Kelvin–Helmholtz instability as revealed by radar Doppler spectra observations

Haoran Li[1], Alexei Korolev[2], and Dmitri Moisseev[1,3]

[1]Institute for Atmospheric and Earth System Research / Physics, Faculty of Science, University of Helsinki, Finland
[2]Environment and Climate Change Canada, Toronto, ON, Canada
[3]Finnish Meteorological Institute, Helsinki, Finland

**Correspondence:** Haoran Li (haoran.li@helsinki.fi)

**Abstract.** Mixed-phase clouds are globally omnipresent and play a major role in the Earth's radiation budget and precipitation formation. The existence of liquid droplets in presence of ice particles is microphysically unstable and depends on a delicate balance of several competing processes. Understanding mechanisms that govern ice initiation and moisture supply are important to understand the life-cycle of such clouds. This study presents observations that reveal the onset of drizzle inside a
∼600 m deep mixed-phase layer embedded in a stratiform precipitation system. Using Doppler spectra analysis, we show how large supercooled liquid droplets are generated in Kelvin-Helmholtz (K-H) instability despite ice particles falling from upper cloud layers. The spectral width of supercooled liquid water mode in radar Doppler spectrum is used to identify a region of increased turbulence. The observations show that large liquid droplets, characterized by reflectivity values larger than -20 dBZ, are generated in this region. In addition to cloud droplets, Doppler spectral analysis reveals the production of the columnar ice crystals in the K-H billows. The modelling study estimates that the concentration of these ice crystals is $3 \sim 8$ L$^{-1}$, which is at least one order of magnitude higher than that of primary ice nucleating particles. Given the detail of the observations, we show that multiple populations of secondary ice particles are generated in regions where larger cloud droplets are produced and not at some constant level within the cloud. It is therefore hypothesized that K-H instability provides conditions favorable for enhanced droplet growth and formation of secondary ice particles.

## 1 Introduction

Clouds strongly influence the Earth's radiation budget (Baker, 1997; Baker and Peter, 2008; Morrison et al., 2012; Tan et al., 2016) and hydrological cycle (Mülmenstädt et al., 2015). A large fraction of clouds are mixed-phase (Hogan et al., 2004), i.e. contain both liquid water droplets and ice particles. Such clouds exist in an unstable equilibrium. Liquid water droplets can rapidly convert into ice particles through Wegener–Bergeron–Findeisen process or riming (Korolev et al., 2017). This transition significantly changes radiative and microphysical properties of these clouds (Sun and Shine, 1994; Lamb and Verlinde, 2011). Climate and numerical weather prediction models, however, struggle in accurately representing mixed-phase clouds (Klein et al., 2009; McCoy et al., 2016; Barrett et al., 2017). They tend to underestimate cloud liquid water content (Klein et al.,



2009; Barrett et al., 2017), which seems to be linked to the modelled ice production (Klein et al., 2009; Barrett et al., 2017). The number of ice crystals is typically controlled by ice nucleating particles (INPs) (DeMott et al., 2010; Kanji et al., 2017). However, in some cases the observed ice crystal number concentration largely exceeds the concentration of INPs (Mossop, 1985; Field et al., 2017). There are several secondary ice production (SIP) mechanisms that explain the production of additional

ice particles (e.g., Hallett and Mossop, 1974; Lauber et al., 2018; Field et al., 2017). Their importance and occurrence, however, is still a topic of the current scientific interest (Field et al., 2017; Korolev et al., 2020; Morrison et al., 2020; Shaw et al., 2020; Luke et al., 2021).

Existence of supercooled liquid water in mixed-phase clouds necessitates sufficient supply of water vapor to replenish its depletion due to ice particles. For single-layered mixed-phase clouds, the water vapor supply may benefit from a number of

processes (Morrison et al., 2012) such as the adiabatic cooling of air parcels due to turbulence (Korolev and Field, 2008) and cloud-scale updrafts (Shupe et al., 2008b), radiative cooling of the liquid cloud layer (Pinto, 1998), entrainment of upper moist air (Solomon et al., 2011) and feedbacks with the surface (Morrison and Pinto, 2006). While multi-layered clouds, where a supercooled liquid water layer is embedded in an ice precipitating cloud, are occurring frequently (Shupe et al., 2006; Intrieri et al., 2002) and may produce drizzle-sized drops (Majewski and French, 2020) as well as secondary ice particles (Hallett and

Mossop, 1974; Lauber et al., 2018; Luke et al., 2021), they are much less studied owing to the complexity of the microphysical processes that are taking place.

Past studies have highlighted the importance of upward air motions, such as shear-induced turbulence (Hill et al., 2014), orographic forcing (Lohmann et al., 2016) and periodic supersaturation variations (Korolev, 1995; Majewski and French, 2020), to the growth of liquid droplets in multi-layered mixed-phase clouds. In addition to these mechanisms, isobaric mixing (Korolev

and Isaac, 2000) and inhomogeneous mixing (Pobanz et al., 1994), usually taking place in dynamically unstable regions such as Kelvin-Helmholtz (K-H) clouds, seem also favor the formation of large supercooled droplets. However, observations with a well depiction of vertical air motions, which are critical for analyzing the driver of the mixing process, seem to be in a shortage (Majewski and French, 2020). Aircraft measurements have been widely utilized in analyzing the growth of liquid drops in mixed-phase clouds (e.g., Korolev and Isaac, 2000; Hogan et al., 2002; Majewski and French, 2020), however they are only

available along the aircraft tracks and cannot resolve the vertical profiles of different hydrometeors. To provide a larger scale view of the clouds, radar observations are often used (e.g., Petre and Verlinde, 2004; Barnes et al., 2018; Luce et al., 2012; Geerts and Miao, 2010; Houser and Bluestein, 2011; Medina and Houze Jr, 2016; Conrick et al., 2018; Grasmick and Geerts, 2020; Gehring et al., 2020). Because radar observations are more sensitive to larger hydrometeors, it is often impossible to separate echoes from ice particles and water droplets at the same time.

In this study, we use dual-polarization Doppler radar spectra observations to separate echoes from cloud droplets and ice particles. The Doppler spectrum as recorded by vertically pointing radars splits radar returns over a range of sampled Doppler velocities (Kollias et al., 2011). The utilization of polarimetric spectral analysis technique (Spek et al., 2008; Moisseev et al., 2015; Luke et al., 2010; Oue et al., 2015; Kalesse et al., 2016; Li and Moisseev, 2020; Luke et al., 2021) allows the identification of supercooled liquid water, which is indicative of vertical air motions, and ice columns, hence facilitates the analysis of

different hydrometeor populations and vertical air motions simultaneously. Using this data and analysis we have uncovered



the K-H billows of supercooled liquid water and secondary ice columns embedded in a stratiform precipitation event. The presented observations advances our understanding of the microphysical processes in stratiform mixed-phase clouds.

The paper is organized as follows. Section 2 introduces the data used in this study. The overview of a stratiform precipitation event is shown in Sect. 3. Section 4 demonstrates the methods used for analysing this event. The dynamics and microphysics

of embedded K-H billows are shown in Sect. 5. Discussions and conclusions are given in Sect. 6 and Sect. 7, respectively.

## 2 Measurements

In this study, measurements from several radars supplemented by radiosonde observations are used to study microphysical and dynamical properties of a stratiform precipitation event that took place on 18 April 2018. The vertically-pointing C- and W-band radars (HYDRA-C and -W) deployed at the University of Helsinki's Hyytiälä Station (Hari and Kulmala, 2005; Petäjä

et al., 2016) (61.845° N, 24.287° E) are aided by a scanning weather radar. This radar setup provides various observing views over Hyytiälä, facilitating the synthetic analysis of clouds and precipitation (Li et al., 2020; Sinclair et al., 2016). The closest sounding station to the Hyytiälä station is located at Jokioinen, which is around 123 km to the southwest of Hyytiälä. The sounding is launched twice a day at around 0 and 12 UTC. The radiosonde observations at 23:30 UTC, the 0 UTC sounding, were used in our analysis

Both vertically pointing radars, HYDRA-C and -W, operate in the linear depolarization ratio (LDR) mode. HYDRA-C is a container-based weather radar that was transported to Hyytiälä in the summer of 2016. Prior to its deployment at the Hyytiälä station, HYDRA-C was operating in Järvenpää (Moisseev et al., 2015). HYDRA-W is a frequency-modulated continuous-wave radar (Küchler et al., 2017) and has been deployed at the Hyytiälä station since November 2017.

The Doppler radar moments, e.g., reflectivity, LDR, Doppler velocity and spectral width, were recorded by both radars. The

pulse length of HYDRA-C is 0.5 $\mu$s, resulting in the range resolution of 75 m, which was oversampled to 50 m. HYDRA-W employs dynamic range resolutions of 25.5 m and 34 m for range gates lower and higher than 3577 m, respectively. The Doppler spectra of HYDRA-W were calculated by applying a Fast Fourier Transform with 1024-point ($V_{\mathrm{max, W}} = 10.24\ m\ s^{-1}$) below and 512-point ($V_{\mathrm{max, W}} = 5.12\ m\ s^{-1}$) above 996 m. The spectral compression mode was used for HYDRA-W and the background noise was removed by applying a noise filter factor which is characterized by the standard deviation of the

Doppler spectrum. The archived W-band polarimetric spectra data enable the analysis of spectral LDR. The time resolutions of HYDRA-C and -W are 1.37 and 3.35 s, respectively. All observations were binned into the time resolution of HYDRA-C and range resolution of HYDRA-W. The beam widths of HYDRA-C and -W are 0.48 ° and 1 °, respectively.

Most of the presented analysis is based on HYDRA-W observations. The C/W dual-wavelength observations at the cloud top were used to compute the path integrated attenuation (Li and Moisseev, 2019; Tridon et al., 2020). This attenuation was

then used to estimate the supercooled liquid water path (SLWP) (Hogan et al., 2005).

The scanning C-band dual-polarization weather radar is located in Ikaalinen (IKA), 64 km west from the Hyytiälä station, and is operated by Finnish Meteorological Institute (FMI). The Ikaalinen C-band radar performs Range-Height Indicator (RHI) scans over Hyytiälä every 15 min. As part of the data analysis, the Ikaalinen radar raw data were processed using the Python



ARM Radar Toolkit (Helmus and Collis, 2016). In this study, the Doppler velocity observations of Ikaalinen radar were employed to identify the radial wind shear over the Hyytiälä station.

Since the Ikaalinen radar is routinely calibrated, its reflectivity measurements were also used to calibrate HYDRA-C. Then, the calibration of HYDRA-W was cross-checked by matching HYDRA-C and HYDRA-W reflectivity values at cloud tops

where the Rayleigh approximation applies at both bands (Hogan et al., 2005; Kneifel et al., 2015; Falconi et al., 2018; Li and Moisseev, 2019; Tridon et al., 2020).

## 3 Overview of the event

A stratiform precipitation system passed over Hyytiälä between 17:00 UTC on 18 April and 04:00 UTC on 19 April 2018. The time-height evolution of this stratiform precipitation captured by HYytiälä Doppler RAdar-W (HYDRA-W, Li and Moisseev,

2020) between 20:00 and 22:30 UTC on 18 April 2018 is presented in Fig. 1a, b and c. As shown in Fig. 1a, the precipitation intensifies after 21:45 UTC. C-band radar reflectivity at 400 m (not shown) increased from $\sim$5 to $\sim$20 dBZ with the rain rate increasing from 0.1 to 0.7 $mm\ h^{-1}$. During the presented time period, the cloud top stays at $\sim$ 4 km, where the air temperature is around -15 °C. The snow generating cells at the cloud top can be identified by the positive Doppler velocities (Fig. 1c). About 1 km below the radar-detected cloud top, the fall streaks visible in the reflectivity observations change their direction, indicating presence of a wind shear layer. It is noteworthy that below the wind shear layer, a region of enhanced spectral width, as can

be seen in Fig. 1b, is present between 21:00 and 21:50 UTC. From 20:20 through 21:20 UTC, the HYDRA-W measured mean Doppler velocity exhibits visible oscillations between 2.3 and 3.2 km (Fig. 1c) implying presence of an embedded convection.

To verify if the environment conditions were favorable for the formation of the K-H instability, the radio sounding data were used. Observations from the radiosonde launched at 23:30 UTC on 18 April 2018 are shown in Fig. 2. The Richardson number

(Ri) was computed both in dry and moist conditions following the method presented in (Hogan et al., 2002). The air between 0.3 and 3.1 km was almost saturated and the moist Ri from 1.7 to 3.5 km was mostly below 0.25, indicating that the necessary condition for development of the K-H instability was met.

Because the radio sounding station is not very close to Hyytiälä, the Doppler velocity observations from Ikaalinen radar were used as additional auxiliary information to support the hypothesis of the formation of K-H instability. From the RHI

measurements vertical profiles of the Doppler velocity observations above the station were extracted. The time-series of these profiles is shown in Fig. 1 (d) and (e). One can observe that IKA radial velocity observations detect a wind shear of 1-2 $m/skm^{-}1$ in the region where the vertical velocity oscillations occur. This wind shear is strongest between 20:40 UTC and 21:40 UTC. It more or less disappears after that. The observed duration of the wind shear is an important observation, which potentially explains the evolution of the K-H billows as will be discussed below. We should also point out that the snow

generating cells, clearly visible in Fig. 1 (a), may affect stability of the layer where we believe the oscillations have formed. This in its turn may affect properties of the K-H instability and it may not have a "classic" appearance. However, the correlation between K-H instability and wind shear, indicates that shear may play a dominant role in formation of the K-H wave.



**Figure 1.** Radar observations collected on 18 April 2018 between 20:00 and 22:30 UTC showing: HYDRA-W reflectivity (a), spectral width (b) and mean Doppler velocity (c). Panels (d) and (e) present vertical profiles of radial Doppler velocity and vertical wind shear derived from the RHI scans of Ikaalinen radar (IKA). Purple and black contours depict the wind shear of -1 and 1 $m\ s^{-1}(km^{-1})$, respectively. The 0°C and -10°C isotherms are marked by gray dashed lines. The black dashed line box shows the region where the spectral analysis has identified presence of supercooled liquid, as shown in Fig. 4.



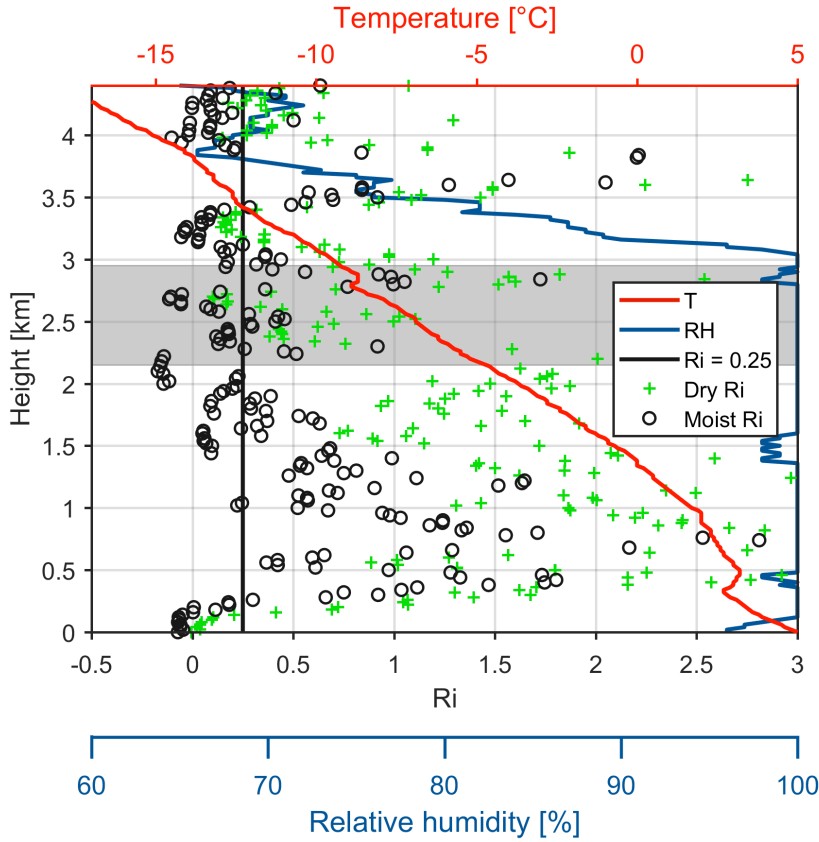

**Figure 2.** Radiosonde observations at 23:30 UTC on 18 April 2018 from the Jokioinen station. This panel shows the air temperature (T, red line), relative humidity (RH, blue line), dry (green cross) and moist (black circle) Ri numbers derived from the radiosonde data. The vertical black line indicates critical Richardson number Ri = 0.25; The gray shading area in this figure corresponds to the heights as marked by black dashed line box in Fig. 1.

## 4   Methods

### 4.1   HYDRA-W Doppler spectra analysis

In a radar volume, hydrometeors with different fall velocities are typically present. Multi-layered mixed-phase clouds is a good example of conditions where hydrometeors with diverse fall velocities are present in a radar volume (Rambukkange et al., 2011; Verlinde et al., 2013; Kalesse et al., 2016). In such cases, analysis of Doppler radar spectra measured by a vertically pointing radar can be used to separate radar echoes of these particles. The sizes of typical supercooled liquid droplets range from 5 to 20 $\mu$m (Shupe et al., 2008b). The terminal velocities are in the order of 0.03 $m\ s^{-1}$ and 0.07 $m\ s^{-1}$ for 10-$\mu$m and 50-$\mu$m liquid droplets, respectively (Kollias et al., 2001). Given their negligible fall velocities, these cloud droplets can be used





as tracers for air motions. In Doppler radar spectra the radar returns from liquid cloud droplets can be identified as a narrow peak around $0\ m\ s^{-1}$. The mean velocity of this peak can be used to derive vertical air motion (Shupe et al., 2004).

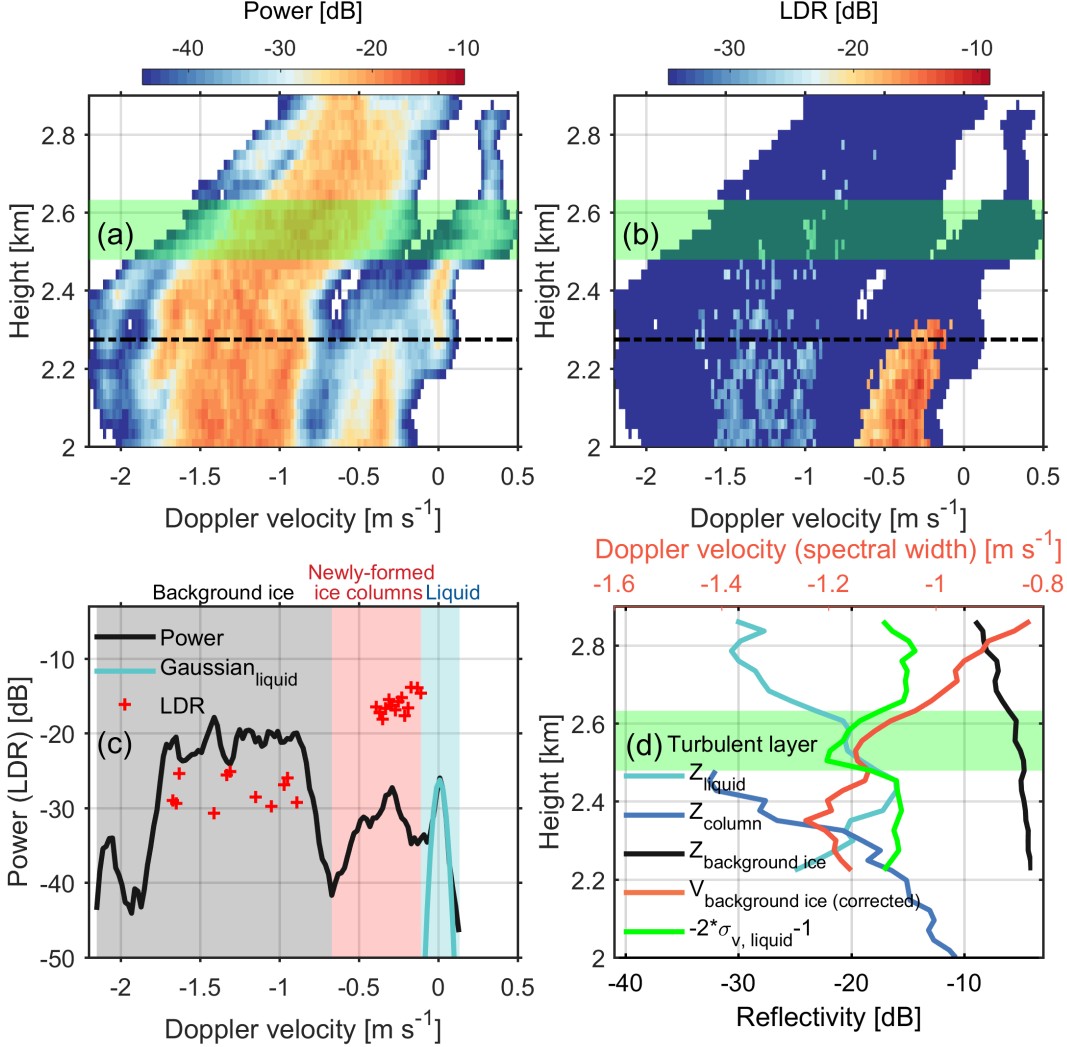

**Figure 3.** Vertical profiles of Doppler spectra at 21:08:43 UTC of (a) power and (b) LDR. (c) A zoomed-in view of the spectral power and LDR as marked by the dot dashed lines in (a, b). (d) Vertical profiles of scaled spectral width of the supercooled liquid water mode (green line) in Doppler spectra ($\sigma_{v,\ liquid}$), Doppler velocity of background ice ($V_{background\ ice}$) after correcting air motions retrieved from liquid water, and radar reflectivity of supercooled liquid water ($Z_{liquid}$), columnar ice ($Z_{column}$) and background ice ($Z_{background\ ice}$). The gray, red and blue shading areas in (c) indicate background ice from above, newly-formed ice columns and supercooled liquid, respectively. The green shading areas in (a, b and d) indicate the turbulent layer as characterized by the increased $\sigma_{v,\ liquid}$. Negative velocity indicates downward.





In multi-layered mixed-phase clouds, more than one population of ice may exist (Zawadzki et al., 2001; Verlinde et al., 2013; Spek et al., 2008). In some cases, the presence of new ice particles is indicative of SIP (Zawadzki et al., 2001). At vertical incidence, LDR can be used to discriminate between columns and other types of ice (Matrosov, 1991; Matrosov et al., 1996; Oue et al., 2015). The columnar crystals will produce LDR values as large as -16 ∼ -13 dB. Figures 3a and b present

vertical profiles of W-band radar Doppler spectral power and LDR, respectively. The supercooled liquid water, with relatively weak and narrow spectral mode (Shupe et al., 2004, 2008b; Luke et al., 2010), exists between 2.2 and 2.85 km. Ice crystals with spectral LDR around -15 dB and Doppler velocity lower than 0.8 $m\ s^{-1}$ are attributed to the columnar ice (Oue et al., 2015). The spectral power of the background ice falling from upper clouds is much larger than that of liquid water droplets and ice columns. A close up to the measurements at the height of 2.28 km is shown in Figure 3c. The background ice, newly-generated

ice columns and supercooled liquid in Doppler spectrum are shaded in gray, yellow and blue, respectively. As can be seen, the spectral LDR of ice columns is much higher than background ice, while the LDR signals for supercooled liquid water are below the noise level.

The Doppler spectra components of different hydrometeor types, present during this event, can be identified by using Doppler velocity and spectral LDR, as discussed above. Despite the straightforward identification of different ice particle types in

Doppler spectrum, deriving their spectral moments, like reflectivity, Doppler velocity and spectral width, is complicated because of the overlap of the spectral modes. This overlap is enhanced if there is significant spectral broadening due to the horizontal and vertical wind shears, and turbulence. As the size distribution of liquid droplets is relatively narrow, their spectrum can be closely approximated by a Gaussian model (Luke and Kollias, 2013). If individual Doppler spectra of different particles can be modelled using Gaussian shapes, then their spectral moments can be estimated by using Gaussian mixture

models (Nguyen et al., 2008). However, the Doppler spectra of ice particles do not necessarily follow this functional form. Nevertheless, the Gaussian model is a good approximation for the Doppler spectrum of liquid cloud droplets (Luke and Kollias, 2013). Since the right slope of the liquid cloud droplet mode in Doppler spectrum is less affected by the ice mode, fitting the Gaussian model can still be done to the right part of the cloud droplet spectrum (Luke and Kollias, 2013). As shown in Figure 3c, the Gaussian model fits the spectral peak of supercooled liquid water well. Using such Gaussian fit, the reflectivity,

spectral width and Doppler velocity of supercooled liquid water were derived. The reflectivity of columnar ice mode in Doppler spectrum was calculated directly from the spectrum, since we expect that the impact of the liquid mode on the reflectivity of columnar ice is not so significant as to affect our study of the evolution of ice columns (Oue et al., 2015; Kalesse et al., 2016).

## 4.2 Estimation of supercooled liquid water path

The retrieval of SLWP was mainly based on the differential attenuation between C- and W-band radar reflectivity (Hogan et al.,

2005).This event was observed by two radars, namely W-band cloud and C-band precipitation radars. While the attenuation due to rain, melting layer, ice particles and supercooled liquid water is negligible for the C-band radar signal, it may have a strong impact on W-band radar observations (Hogan et al., 2005; Li and Moisseev, 2019). The differential attenuation between C- and W-band radar observations caused by supercooled liquid water is proportional to the liquid water content (Hogan et al., 2005).



Therefore, if the attenuation due to other attenuation sources can be mitigated, then the differential attenuation measurements can be used to estimate SLWP.

For HYDRA-W, the observed reflectivity is affected by the attenuation from atmosphere and wet radome. Using the reflectivity observations from HYDRA-C, the atmospheric attenuation due to rain and melting layer at W-band was removed by

applying reflectivity - attenuation relations (Li and Moisseev, 2019). It was found that the attenuation due to rain and melting layer is relatively small, less than 1 dB, and the uncertainties in attenuation estimation were in the order of 0.5 dB (Li and Moisseev, 2019). Given the weak rain intensity, the attenuation due to ice particles is negligible (Leinonen et al., 2011). The gaseous attenuation can be calculated from the millimeter-wave propagation model (Liebe, 1985). The wet radome attenuation should be minimized since the antenna of HYDRA-W is protected by a hydrophobic cover and the blower further reduces

the radome wetting. After the attenuation due to rain and melting layer was removed from W-band reflectivity, the differential attenuation was derived by matching C- and W-band reflectivities at the cloud top, where ice particles are expected to be small enough to satisfy the Rayleigh approximation in both radar bands. Then, the SLWP was converted from the differential attenuation which is mainly caused by the liquid attenuation at W-band (Hogan et al., 2005).

## 5 The Kelvin–Helmholtz billows

The following discussion is focused on the analysis of dynamics and microphysical properties of the cloud region inside the black rectangle in Fig. 1 a,b,c (20:50 $\sim$ 21:40 UTC). This period will be analyzed with the help of specifically derived radar reflectivity ($Z_{\text{liquid}}$), spectral width ($\sigma_{\text{v, liquid}}$) and Doppler velocity of the supercooled liquid water ($V_{\text{liquid}}$), reflectivity of ice columns ($Z_{\text{column}}$), and SLWP. It should be noted that the rectangle in the figures depicts the region where the supercooled liquid water was detected. The Doppler analysis was also performed on the regions outside of the rectangle, but no supercooled

water or columnar ice production was detected there.

### 5.1 Doppler velocity and spectral width of supercooled liquid water

A layer of supercooled liquid water persisted during the entire period (Fig. 4). Since the cloud droplet fall velocity usually does not exceed few $cm\ s^{-1}$ ($\sim$0.03 $m\ s^{-1}$ for the diameter of 10 $\mu$m), $V_{\text{liquid}}$ is used for the assessment of vertical velocity of air motion (Shupe et al., 2004). Fig. 4a reveals periodic air circulations with the amplitude of approximately 0.4 $m\ s^{-1}$. It should

be noted that the K-H instability may not be the only factor modulating air motions, since the wind shear layer is so close to the cloud top. However, we argue that the oscillation of air motions at around 2.8 km is mainly caused by the K-H instability. Firstly, the wind shear layer, which is inductive to vertical air motions, at 2.8 km was clearly present from 20:50 to 21:40 UTC as shown in Fig. 1e. Secondly, after 22:00 UTC where the wind shear layer has almost disappeared, there are no identifiable signatures of air oscillations at 2.8 km (Fig. 1c).

A wave train of enhanced $\sigma_{\text{v, liquid}}$ at around 2.5 km is collocated with vertical air perturbations (Fig. 4b). Specifically, the upward branches of the waves coincide with updrafts, and the crests of waves are close to the transitional regions of upward and downward velocities. Such collocation of K-H billows and vertical air motions bears a good resemblance to previous vertically-

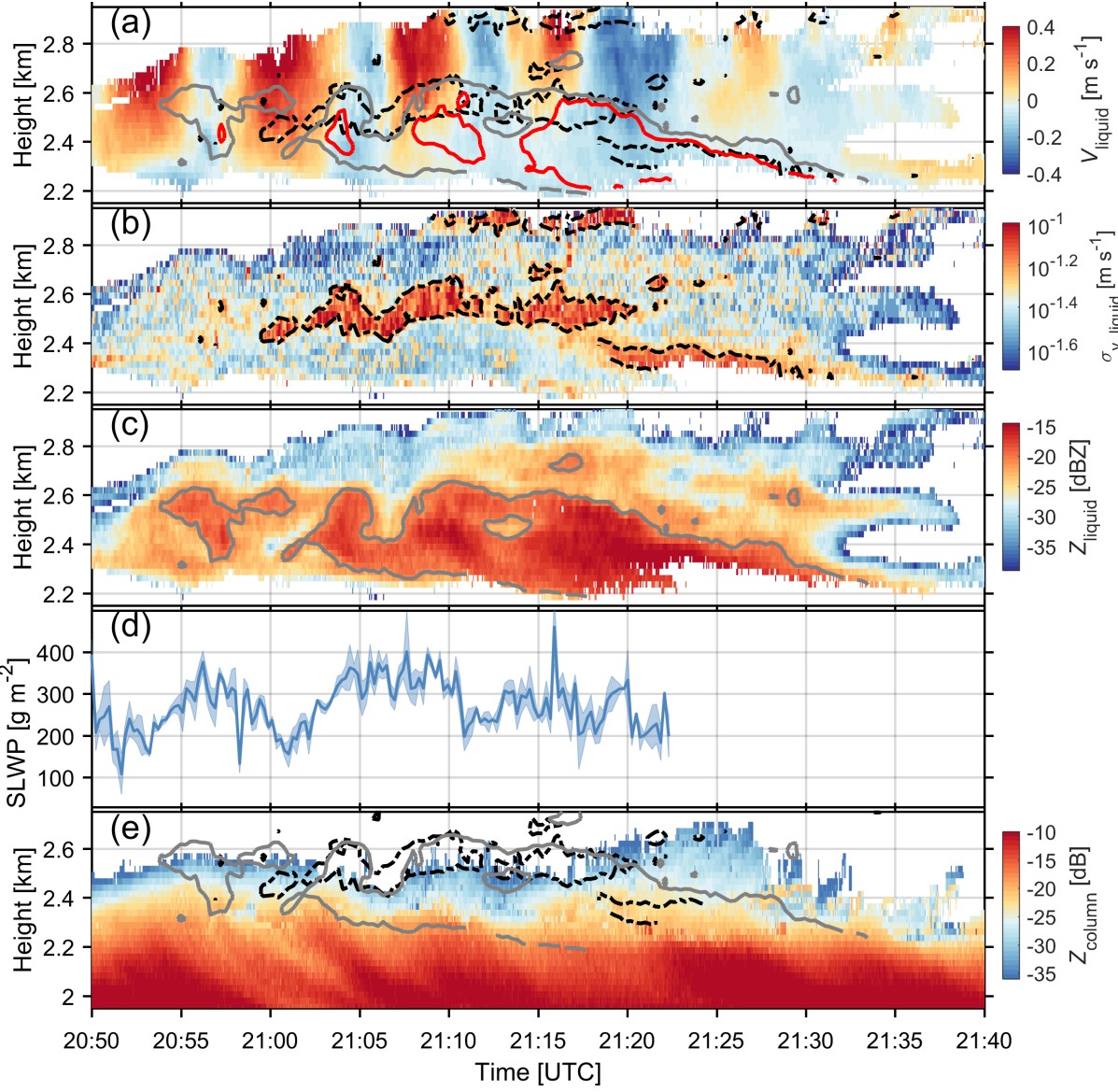

**Figure 4.** (a) Doppler velocity, (b) spectral width and (c) reflectivity of supercooled liquid cloud droplets, (d) SLWP above melting layer estimated from the differential attenuation between C- and W-bands, (e) reflectivity of ice columns. Red and gray isolines indicate respectively the reflectivity of -18 and -21 dBZ for supercooled liquid water. Black dashed isolines indicate the spectral width of 0.06 $m\ s^{-1}$ for supercooled liquid water. The shading areas in (d) indicate the standard deviation of SLWP in 13.7 s. After 21:23 UTC, no SLWP retrieval was made. This was caused by the rapid changes of the observed C-band reflectivity at cloud top, which hindered the identification of Rayleigh scattering regions.





pointing radar records of K-H clouds (Petre and Verlinde, 2004; Geerts and Miao, 2010; Luce et al., 2012). Despite the lower amplitude of the vertical velocities in the present observations, both temporal ($\sim$ 5 min) and spatial ($\sim$ 300 m) scales of each wave are very similar to those K-H billows observed in the convective outflow anvil (Petre and Verlinde, 2004). In addition, $\sigma_{\text{v, liquid}}$ is enhanced along the central axis (Barnes et al., 2018) rather than only around the crests of K-H waves (Houser and

Bluestein, 2011).

The enhanced $\sigma_{\text{v, liquid}}$ can be caused by the broadening of the drop size distribution as well as the enhanced turbulence within the radar volume. If it is attributed to the former, the increase of $\sigma_{\text{v, liquid}}$ would not be just 100 to 200 m in depth and vanish just below 2.4 km. Therefore, we conclude that the sharp increase of $\sigma_{\text{v, liquid}}$ iss mainly caused by the velocity fluctuations, and mixing was taking place between 2.4 km and 2.6 km.

**5.2   Reflectivity of supercooled liquid water**

The regions of enhanced $Z_{\text{liquid}}$ manifest themselves as well-defined K-H billows (see gray solid isolines in Fig. 4c), especially between 21:00 and 21:13 UTC. The crests of the K-H billows are around the transitional regions of upward and downward velocities and protrude to downdrafts. Just below the turbulent layer as characterized by the increased $\sigma_{\text{v, liquid}}$, there is a region of enhanced $Z_{\text{liquid}}$ which is as high as -16 dBZ, indicating onset of drizzle in this region (Frisch et al., 1995).

Interestingly, the K-H billows of $Z_{\text{liquid}}$ (gray solid isolines) are collocated with the enhanced $\sigma_{\text{v, liquid}}$ (black dashed isolines) as shown in Fig. 4a. This collocation resembles what has been observed in rain where the perturbation from the K-H instability is expected to facilitate the coalescence of raindrops and leads to increased radar reflectivity (Barnes et al., 2018).

Previous observational studies have shown that K-H billows embedded in precipitation can be characterized by the fluctuations of radar reflectivity and spectral width collocating with Doppler velocities (Petre and Verlinde, 2004; Geerts and Miao,

2010; Barnes et al., 2018). In this event, such signatures are hardly identifiable in conventional radar reflectivity and spectral width observations (Fig. 1a and b), despite the evidence of air circulations as indicated by mean Doppler velocities (Fig. 1c).

**5.3   Supercooled liquid water path**

As shown in Fig. 4d, the SLWP presents a wave-like pattern as well. The estimated SLWP ranges from 150 to 400 $g\ m^{-2}$ between 21:00 and 21:15 UTC. In general, the SLWP is larger in downdrafts than updrafts, somewhat resembling aircraft

observations presented in (Majewski and French, 2020). The oscillations in SLWP and observed supercooled liquid droplet velocities do not fully coincide. This discrepancy can be attributed to presence of supercooled liquid at the other altitudes, i.e. at around 1.5 km (not shown) and potentially at the top of the snow generating cells.

**5.4   Generation of ice columns**

$Z_{\text{column}}$ was derived from the columnar ice mode in radar Doppler spectrum observations. As shown in Fig. 4e, the detected ice

columns initiate at about 2.5 km, coinciding with the enhanced $\sigma_{\text{v, liquid}}$. Once generated, these ice particles grow rapidly from about -30 dBZ to -15 dBZ within $\sim$0.3 km. Interestingly, several fall streaks of ice columns present below the K-H billows,





and each of them (around 21:04, 21:10 and 21:22 UTC, respectively) corresponds to a crest in the K-H billows as indicated by the enhanced $Z_{\mathrm{liquid}}$ and $\sigma_{\mathrm{v, liquid}}$.

## 5.5 Multiple populations of ice columns

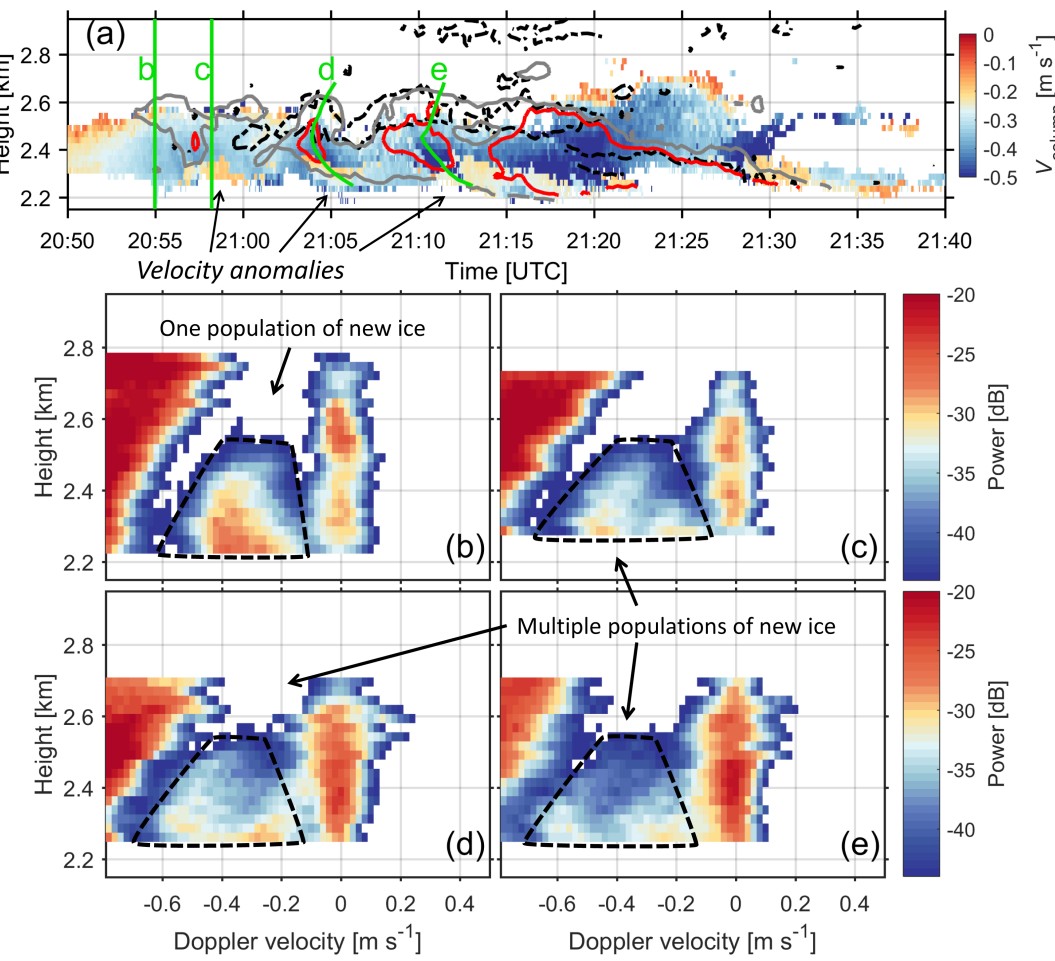

**Figure 5.** (a) Doppler velocity of ice columns after the air motion correction. For green lines b, c, d and e, the corresponding Doppler spectral power observations after the air motion correction are shown in (b), (c), (d) and (e), respectively. Red and gray isolines in (a) indicate respectively the reflectivity of -18 and -21 dBZ for supercooled liquid water. Black dashed isolines in (a) indicate the spectral width of 0.06 $m\ s^{-1}$ for supercooled liquid water. Black arrows point to the regions of velocity anomalies in (a). Populations of new ice in (b), (c), (d) and (e) are marked by black dashed curves.

Based on the air motions estimated from the $V_{\mathrm{liquid}}$, we have been able to estimate the terminal Doppler velocity of ice
5 columns ($V_{\mathrm{column}}$) as shown in Fig. 5a. Before 20:56 UTC, $V_{\mathrm{column}}$ increases as ice columns fall toward the ground. This is expected, since the mass of ice columns increases during riming (if they are not too small) and depositional growth, leading to





increased terminal velocity (Lamb and Verlinde, 2011). After 20:56 UTC, intermittent slowdown of $V_{\text{column}}$ at around 2.3 km occurred (black arrows). This anomalous decrease of $V_{\text{column}}$ is surprising, given the fact that the air motion has already been corrected. In addition, the velocity anomalies are well collocated with the areas of high $Z_{\text{liquid}}$ (above -18 dBZ, red isolines).

Further inspection was made into the radar Doppler spectra observations. Fig. 5 b, c, d and e show the observed Doppler spectral power of the green lines b, c, d and e in Fig. 5 a, respectively. The green lines d and e were identified by tracing the peaks of $Z_{\text{liquid}}$. It is anticipated that the green lines are representative of the quasi-Lagrangian trajectories of ice columns. As expected, there seems to be one population of ice columns in Fig. 5b, which corresponds to the period with no velocity anomaly. In contrast, multiple populations of ice columns can be identified in Fig. 5 c, d and e. At the height of 2.3 km, the velocity of faster falling columnar ice is $0.5 \sim 0.6 \; m \; s^{-1}$ while the velocity of slower falling ones is $0.2 \sim 0.3 \; m \; s^{-1}$. Fig. 5 d and e suggest that the generation of the faster falling ice columns coincides with the turbulent layer at around 2.55 km whereas the slower ones were formed at around 2.4 km which corresponds to the region of enhanced $Z_{\text{liquid}}$.

## 6    Discussions

### 6.1    Generation of large droplets

As shown in Fig. 4c, the reflectivity of supercooled liquid water can be as high as -18 $\sim$ -15 dBZ, which exceeds (or is close to) the expected reflectivity of drizzle, e.g., -20 dBZ (Kato et al., 2001), -17 dBZ (Kogan et al., 2005), and -15 dBZ (Chin et al., 2000). Generally, the reflectivity of liquid clouds seldom exceeds -18 dBZ (Frisch et al., 1995). This, therefore, indicates onset of drizzle that is taking place in the K-H billows.

One of the striking observations is that the level where velocity oscillations are most intensive is obviously higher than the regions of enhanced reflectivity of supercooled liquid water. This suggests that the droplet growth at quasi-steady supersaturation formed in adiabatically ascending cloud is not the only mechansims responsible for the drizzle formation (Houze Jr and Medina, 2005; Shupe et al., 2008b; Houser and Bluestein, 2011; Morrison et al., 2012). Given the apparent presence of enhanced turbulence (Fig. 4b), we speculate that the drizzle formation could be associated with isobaric mixing. The supersaturation caused by isobaric mixing of saturated air parcels with different temperatures can facilitate the generation of drizzle (Korolev and Isaac, 2000). Fig. 2 shows presence of the inversion around 1.5 °C embedded in the liquid layer at $\sim$2.75 km. Vertical transport of cloud parcels forced by the wind shear to the levels below and above the inversion will result in temperature fluctuations in a saturated environment required for the generation of cloud volumes with high supersaturation formed due to isobaric mixing. The mixing process is supported by the enhanced spectral width (Fig. 4b) which is indicative of strong sub-radar-volume velocity fluctuations. The collocation of enhanced spectral width and reflectivity further indicates the strong link between the mixing and large drop formation. The relatively high SLWP values during downdrafts seem further corroborating this hypothesis. The temperature fluctuation of 2.5 °C can lead to a supersaturation of 0.5% (Korolev and Isaac, 2000) which is favorable for the generation and growth of supercooled liquid droplets. Though the observed inversion over the radiosonde station is relatively weak and the temperature difference between the airs below and above the inversion is around 1.5 °C, we speculate that the actual inversion over the Hyytiälä station might be stronger.



The inhomogeneous mixing in the K-H instability can also contribute to the generation of large supercooled droplets (Pobanz et al., 1994). As soon as the liquid droplets meet with dry air, they evaporate until the saturation is reached or running out of the liquid water. This mixing process facilitates the formation of large droplets thanks to the depletion of small liquid droplets hence more excess water vapor is available for growing large cloud droplets (Lamb and Verlinde, 2011). The circulation of this

process allows the continuous growth of liquid droplets (Korolev, 1995) as supported by recent aircraft observations (Majewski and French, 2020). After overcoming the mechanistic gap around the diameter of 40 $\mu m$ (Lamb and Verlinde, 2011), these liquid drops grow more efficiently during the collision-coalescence process (Korolev and Isaac, 2000).

## 6.2 Secondary ice production

As follows from Fig. 4, the first radar echo of ice columns was detected at approximately 2.5 $\sim$ 2.6 km. Since ice crystals

must grow for some time in order to become detectable by the radar, the actual level of the columns' origin is most likely located at the top of the mixed phase layer at 2.65 km. The temperature at this altitude was around -8 °C, which is expected for the columnar growth. During their fall through the mixed-phase layer, the columns grew and the average fall velocity reached approximately 0.35 $m\ s^{-1}$ as derived from the Doppler spectrum at 2.2 km. To assess the back trajectory of ice crystals, a simulation of free-falling growing ice columns was performed. In this simulation, the cloud environment was considered to be

saturated over liquid water.

Figure 6 shows modelled changes of mass, maximum length ($L_{\mathrm{max}}$) and $V_{\mathrm{column}}$ between 2.2 km and 2.7 km for columnar crystals with aspect ratios of 2, 4 and 8. The radar-observed $V_{\mathrm{column}}$ between 21:08:20 and 21:08:50 UTC where one population of ice columns presented was used for the assessment (black curve in Fig. 6c). In order to get the simulated $V_{\mathrm{column}}$ consistent with radar measurements, the aspect ratio (AR) of ice columns formed at 2.6 $\sim$ 2.7 km should be 4 $\sim$ 8, which implies the mass

of these columns was 2 $\sim$ 3 $\times$ 10$^{-6}$ g at around 2.2 km. At this level, $Z_{\mathrm{column}}$ (Fig. 3d) is -15 $\sim$ -18 dB. Hence, the ice water content (IWC) of ice columns was 0.009 $\sim$ 0.014 $g\ m^{-3}$ based on the parameterization IWC = 0.167Z$^{0.712}$ which was derived using aircraft observations within the temperature of -12 $\sim$ -3 °C (Korolev et al., 2020). Therefore, the number concentration of ice columns ($N = mass/IWC$) is in the range of 3 $\sim$ 8 L$^{-1}$. Interestingly, the $N$ estimated from the temperature-dependent $IWC - Z$ parameterization (Hogan et al., 2006) is within the same magnitude (4.7$\pm$3.7 L$^{-1}$ for $Z_{\mathrm{column}}$ of -10 dB at 2 km, -5

°C). The estimated primary INPs based on Demott et al. (DeMott et al., 2010) parameterization is about 0.3 L$^{-1}$ at -8 °C. The INP parameterization (Schneider et al., 2021) derived from measurements obtained at the Hyytiälä station yields the primary INP concentration of 7 $\times$ 10$^{-4}$ L$^{-1}$ at -8 °C. These values are about 1 $\sim$ 5 orders of magnitude lower than the estimated ice number concentration. Therefore, the estimated concentration of columns cannot be explained by the primary nucleation. In addition, their origin in the mixed-phase layer cannot be explained by the seeding ice from above, since the columnar shape is

not consistent with the environmental conditions in the cloud above. As a result, the SIP is the most likely explanation for the observed columnar ice. Considering that smaller ice columns contribute less to the radar reflectivity, the actual concentration of columns is expected to be even higher than the estimation. This provides an additional support to the ice initiation through SIP.





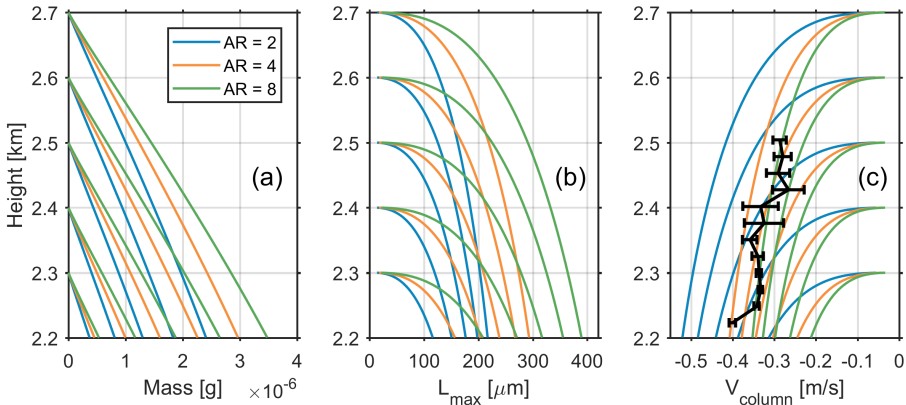

**Figure 6.** Simulated (a) mass, (b) maximum length ($L_{\text{max}}$) and (c) fall velocity ($V_{\text{column}}$) of ice columns with aspect ratios (AR) of 2, 4 and 8. $V_{\text{column}}$ was calculated based on Kajikawa Kajikawa (1976) parameterization. Ice particles were considered to grow at saturation over liquid water. Observations from the radiosonde launched at 23:30 UTC were used as inputs. The radar-observed $V_{\text{column}}$ (black) from 21:08:20 UTC to 21:08:50 UTC is shown in (c) and the error bars indicate the standard deviation of $V_{\text{column}}$ during this period.

There are two potential mechanisms to explain the SIP in our observations: the Hallett-Mossop (H-M) process (Hallett and Mossop, 1974; Mossop and Hallett, 1974) and droplet breakup during freezing (Field et al., 2017; Lauber et al., 2018; Keinert et al., 2020; Lauber et al., 2021). Activation of the H-M process requires a set of necessary conditions: (a) the presence of droplets larger than 24 $\mu$m; (b) environmental temperature of -8 $\sim$ -3 °C and (c) the presence of rimed snow flakes. As

5    follows from the above discussion, conditions (a) and (b) are satisfied in the mixed-phase layer. The presence of graupel and heavily rimed ice is supported by the observed Doppler velocity (around 2 $m\ s^{-1}$ after compensating for air motions, Fig. 3 a). Therefore, the environmental conditions required for the initiation of the H-M process were satisfied in the mixed-phase layer. The presence of drizzle-sized drops and the background ice is also favorable for SIP via the droplet breakup during freezing (Lauber et al., 2018; Keinert et al., 2020). This is very possible, given Luke et al. (2021) have recently proposed that the droplet

10    breakup is more efficient than H-M process in natural clouds. The background ice particles in this case can play the role of INPs. Hence, it is concluded that both mechanisms may be responsible for the observed SIP.

For the first time, multiple populations of newly generated ice particles have been identified in K-H billows. The spectral analysis method presented in study shows the potential of remote sensing techniques in studying the evolution of SIP event. Particularly, although the spatially separated SIP cloud regions can be identified from the "needle-like" aircraft penetrations

15    (e.g., Hogan et al., 2002; Korolev et al., 2020), our understanding on their distribution is rather limited (Field et al., 2017). Summarizing the obtained observations, it could be stated that K-H instability embedded in a relatively deep mixed-phase layer may create conditions favorable for enhanced growth of large droplets, which facilitate the initiation of SIP. Several past observations suggest that, in many cases,the SIP was spatially associated with dynamically active cloud regions (Korolev et al., 2020; Lasher-Trapp et al., 2016; Rauber et al., 2019). The present study supports this previously attained finding. As





a result, we advocate that more efforts should be devoted to addressing the microphysical impact of SIP, as currently poorly parameterized in models (Morrison et al., 2020), associated with the K-H instability.

## 7    Conclusions

On 18 April 2014 between 20:50 and 21:40 UTC, a K-H cloud embedded in stratiform precipitation was observed over the
Univerisity of Helsinki measurement site in Hyytiälä, Finland. The observation were collected using three radars, two vertically pointing C- and W band radars located at the site and a scanning weather radar operated by FMI. The FMI radar is located 64 km west of the site and was performing RHI scans every 15 min over Hyytiälä. Although oscillations of mean vertical Doppler velocity are visible in the data, no obvious signatures of K-H waves were identified from radar reflectivity and spectral width observations.

To unmask the cloud, radar Doppler spectra analysis was performed on the W-band radar observations. The analysis showed presence of supercooled cloud embedded in the precipitation. The mean Doppler velocities of the cloud droplets, which can be used as tracers of air motion, exhibited well defined oscillation with the magnitude of 0.4 m/s. The FMI weather radar observations detected presence of vertical shear layer coinciding with the supercooled liquid cloud layer. Given that the life-time of the cloud coincides with the life-time of the shear, we speculate that this cloud is mainly formed due to the K-H
instability. It should be noted, however, that other processes could also contribute to the formation of the atmospheric wave, i.e. air motion associated with the snow generating cells that are present about 0.5 km above the studied layer.

The presented observations and analysis show that K-H clouds are capable of producing large liquid droplets, despite of the competition for the moisture with ice particles falling above. In some regions, the observed K-H cloud reflectivity values exceeded -20 dBZ, which is typically associated with onset of drizzle. Furthermore, spectral linear depolarization ratio obser-
vations identified presence of ice columns. These columns appeared to form in the supercooled liquid cloud regions. In the pockets where larger cloud droplets were formed, multiple populations of columnar ice particles were observed. The estimated number concentration of ice columns is at least by one order of magnitude higher than the expected concentration of primary ice-nucleating particles. This implies that the secondary ice production is the most likely source of the observed columnar ice.

Overall the presented observations and analysis indicate that even not very strong wind shears may result in formation of
K-H instability, which could lead to formation of conditions favorable for onset of supercooled drizzle and secondary ice production.

*Data availability.* Radiosonde observations can be accessed at http://weather.uwyo.edu/upperair/sounding.html. The radar data used in this study are available at http://doi.org/10.5281/zenodo.4019602.





*Author contributions.*  HL and DM conceptualized the study and wrote the manuscript. HL performed the majority of the data analysis. All the authors took part in the interpretation of the results. AK performed the ice particle growth modeling, reviewed and edited the manuscript.

*Competing interests.*  The authors declare that they have no conflict of interest.

*Acknowledgements.*  We thank Matti Leskinen for his excellent work on the instrument maintenance. Jussi Tiira is acknowledged for his help
5   in processing radar data. This research has been supported by the Horizon 2020 (grant nos. ACTRIS-2 654109, ACTRIS PPP 739530, ACTRIS-IMP 871115, ATMO-ACCESS 101008004, ERA-PLANET iCUPE, 689443) and Academy of Finland (ACTRIS-NF 328616, ACTRIS-CF 329274 and Center of Excellence in Atmospheric Sciences, 307331, Atmosphere and Climate Competence Center, 337549), University of Helsinki (ACTRIS-HY). Haoran Li was also funded by China Scholarship Council.



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
