# Peer review of "Supercooled liquid water and secondary ice production in Kelvin–Helmholtz instability as revealed by radar Doppler spectra observations"

_Atmospheric Chemistry and Physics, 2021_

## Author Comment (AC1)

**Reviewer #1**

The authors investigated the origins of supercooled liquid water and secondary ice in a stratiform precipitation event. Vertically pointing C-band and W-band radars at the Hyytiälä station, scanning C-band weather radar at Ikaalinen, and sounding at Jokioinen, provide rich information of the microphysical and dynamical properties of the mixed-phase layer embedded in the stratiform precipitation system. Radar doppler moments (reflectivity, linear depolarization ratio, Doppler velocity and spectral width) are used to explore cloud dynamics (turbulent layer and wind shear) and identify various hydrometer types (including supercooled cloud droplets, newly-formed ice columns, and background ice). Path integrated attenuation from the C/W dual-wavelength observations also provide an estimation of supercooled liquid water path. Valuable data and careful analysis provide physical insights of the formation of supercooled liquid water and secondary ice production due to a turbulent layer caused by Kelvin-Helmholtz instability: (1) Shear leads to the formation of the Kelvin-Helmholtz instability; (2) Drizzle forms in the K-H billows due to isobaric mixing; (3) Secondary ice forms in the K-H billows due to either the Hallett-Mossop process or droplet breakup during freezing, or both. The manuscript is well written, and results are quite convincing. I suggest the publication of this paper in ACP with only some minor comments below:

We sincerely appreciate the reviewer for the positive comments on our paper. We have amended the manuscript as suggested. Please see below our response to your comments.

1. Figure 4: Blank regions in the Figure (e.g., above 2.8 km at 20:55 UTC) are when liquid cloud droplet mode does not exist in Doppler spectrum? Please state it clearly in the text or caption.

   Thank you for the very good suggestion. We have added the following text in the caption:

*Blank regions in (a, b, c and e) are where no significant supercooled liquid mode (a, b, c) or columnar ice mode (e) can be identified in radar Doppler spectrum.*

2. Page 14, Line 13-14: "To assess the back trajectory of ice crystals, a simulation of free-falling growing ice columns was performed." Please provide more details (e.g., literature, equations) of the ice particle growth model.

   In the revised manuscript, we have added the information about the ice particle growth model in appendix.

3. Page 14, Line 23: "N=mass/IWC". Should it be "N=IWC/mass"?

   Corrected.

---

## Author Comment (AC2)

**Reviewer #2**

Using observations by vertical pointing C- and W-band radars, scanning weather radar, and radiosondes, the authors analyzed dynamics and microphysics of supercooled liquid water and secondary ice in a stratiform drizzling cloud. It is a unique and interesting case that K-H cloud has developed in the stratiform cloud. The authors point out that the K-H instability is induced mainly by wind shear. They also revealed that the number concentration of ice columns is higher than the INP concentration by several degrees of magnitude, which indicates the secondary ice production in the K-H billows.

The manuscript is well written, the subject is relevant, the observation data is most advanced, and the results are well presented and discussed. I believe the manuscript is suitable for publication after a minor revision.

We sincerely appreciate the reviewer for the positive comments on our paper. We regard these very detailed comments as valuable suggestions which significantly improved the readability this manuscript. Therefore, we have amended the manuscript following almost all the suggestions, while we tend to keep the original version in some places after very careful consideration. Please see below our response to your comments.

Minor points:

1. Page 2, line 30: The description of the dual-polarization Doppler radar technique should go to the data and methodology section, and what do you find from this study should go to the discussion or conclusion sections. The content of this paragraph does not fit the "introduction".

   We have amended the introduction section as suggested.

2. Page 4, line 20: (Hogan et al., 2002) -> Hogan et al. (2002).
   Corrected.

3. Figure 1: The text, for example, "Vertically pointing (HYDRA-W)", is not visible.

   Amended.

4. Figure 2: a potential temperature profile in Kelvin other than temperature is better to interpret the stability in the boundary layer.

   In this Figure, we use sounding observations as auxiliary evidence to support the existence of KH instability, similar with Hogan et al., (2002). Hence, we

want to focus on Ri instead of explaining the static instability. Therefore, we have decided to keep this figure concise.

5. Figure 4: use the same y-axis limits for heights in (a), (b), (c), and (e).

We agree that the same y-axis makes the figure look more friendly. However, we want to focus on the most interesting region in each subfigure. For supercooled liquid water, it is 2.2 ~ 2.9 km, while significant columnar ice production appears below 2.6 km. To mitigate the discordance of y-axis limits of (a,b,c) and (e), we have added isolines of reflectivity, which help to identify the relative position between KH clouds and ice columns. Therefore, we have decided to keep the original layout of this figure.

6. Page 11, line 9: iss -> is.
Corrected.

7. Page 11, line 18-21: Could you explain why your results are different from previous studies?

We have added the following texts after Line 21.

*This difference may be explained by much weaker vertical air motions and the potential impact of snow generating cells at cloud top in this study.*

8. Page 11, line 25: (Majewski and French, 2020) -> Majewski and French (2020).

Corrected.

9. Page 14, line 23: What is "mass" in the equation "N = mass/IWC", and double-check if the equation is correct.

Corrected.

10. Page 16, line 6: "W band" -> "W-band".

Corrected.